# Topographic changes in macula and its association with visual outcomes in idiopathic epiretinal membrane surgery

**Seung Min Lee**[1,3], **Sung Who Park**[2,3], **Iksoo Byon**[2,3] *

1 Department of Ophthalmology, Research Institute for Convergence of Biomedical Science and Technology, Pusan National University Yangsan Hospital, Yangsan, South Korea, 2 Department of Ophthalmology, Medical Research Institute, Pusan National University Hospital, Busan, South Korea, 3 Pusan National University School of Medicine, Yangsan, South Korea

* isbyon@naver.com, isbyon@pusan.ac.kr

## Abstract

### Purpose

We investigated changes in macular topography and their association with visual acuity and metamorphopsia in the idiopathic epiretinal membrane (iERM).

### Methods

Twenty-four eyes that underwent vitrectomy and ERM removal with internal limiting membrane peeling were included in this study. Best-corrected visual acuity (BCVA) and horizontal/vertical metamorphopsia scores (h and vM-scores in the M-chart) were assessed. The distances of fovea-disc (FD) and fovea-vascular arcade (FV), central subfield macular thickness (CSMT), and foveal location were measured using fundus photography, optical coherence tomography (OCT), and OCT angiography, respectively.

### Results

The BCVA (logMAR) and vM-scores ($P < 0.001$, $P = 0.014$, respectively) improved after surgery. The distance of the FD decreased ($P < 0.001$) and FVs increased ($P < 0.001$, both). The fovea moved horizontally toward the disc ($P < 0.001$). The change in median total FVs (7.114 mm to 7.369 mm, $P = 0.001$) correlated with the change in BCVA ($P = 0.049$, Pearson's [r] = -0.404). No topographic parameters were associated with an improvement in the vM score.

### Conclusions

The macular topography significantly changed after iERM removal; the fovea moved nasally, and the distance between the superior and inferior vascular arcades increased. Such a change was relevant to the improvement in BCVA, but not metamorphopsia.

**Data Availability Statement:** All relevant data are within the manuscript and its Supporting information files.

**Funding:** The author(s) received no specific funding for this work.

**Competing interests:** The authors have declared that no competing interests exist.

## Introduction

Idiopathic epiretinal membrane (iERM) commonly appears in the older population as a glistening amorphous membrane on the macula [1–3]. The membrane is composed of glial cells migrating through the inner limiting membrane by posterior vitreous detachment (PVD) or hyalocytes with anomalous PVD and extracellular matrix [4–8]. These cells acquire contractibility by myofibroblastic transdifferentiation, deforming the macula [8, 9]. The gradual distortion of the retinal layers results in visual impairment, including metamorphopsia and visual loss. Vitrectomy and membrane removal are recommended in symptomatic patients [10, 11].

Many factors influence visual outcomes in iERM. Baseline VA is one well-known factor [3]. Anatomical factors that predict visual outcomes have been studied in various imaging assessments. Serial fundus photographs showed both the membrane reflex on the macula in the early stage and the wrinkled, opaque macula with decreased distance between vascular arcades at an advanced stage [2]. High resolution images from optical coherence tomography (OCT) have provided detailed anatomic factors associated with visual outcomes: the photoreceptor integrity, inner nuclear layer (INL) thickness, ectopic inner foveal layer (EIFL), and macular thickness [12–17]. OCT angiography (OCTA) noninvasively generates images of micro-vasculatures in the macula, and precisely localizes the foveal avascular area. Many studies using OCTA reported that a small and irregular fovea avascular zone was a poor prognostic factor [18, 19].

However, previous studies did not used multimodal imaging to assess various prognostic factors. Various factors influence each other in eyes with iERM, which can be confounding factors in predicting prognosis. Previously studied methodologies failed to identify confounding variables that might affect prognosis of surgery [13, 15]. For example, an increase in macular thickness is highly associated with a decrease in the distance between vascular arcades. Therefore, the prognostic value of both the thickness and distance must be simultaneously assessed using imaging devices. Multimodal imaging can provide more accurate information on the association between structural changes and visual function in iERM. Here, we investigated changes in macular topography and their association with visual acuity and metamorphopsia using multimodal imaging analysis of serial fundus photographs, OCT, and OCTA in iERM.

## Materials and methods

### Study participants

We retrospectively reviewed the data of consecutive patients diagnosed with unilateral iERM who underwent pars plana vitrectomy and membrane removal by a single surgeon (I.B.) at the vitreoretinal clinic of Pusan National University Yangsan Hospital, South Korea from December 1, 2015 to May 31, 2017. iERM was diagnosed based on the presence of a thin membrane on the surface of the retina on swept-source OCT images (DRI OCT-1 Atlantis, Topcon Corp., Tokyo, Japan) and a fibrous membrane in front of the macula on fundus examination.

This study included patients who had intact ellipsoid zones (EZ) of the photoreceptor layers and were followed up for at least 6 months after iERM removal. Eyes presenting with an idiopathic epiretinal membrane that typically covered the fovea without features such as lamellar hole, pseudohole, or vitreomacular traction syndrome were included. Patients were excluded if they exhibited (1) presence of a secondary ERM due to retinal break, uveitis, diabetic retinopathy, retinal vascular disease, trauma, intraocular infection or history of photocoagulation; (2) high myopia (spherical equivalent $\geq$ -6.0 diopters or axial length $\geq$ 26 mm); (3) previous intraocular surgery with the exclusion of uncomplicated phacoemulsification; (4) media opacities that hinder the evaluation of the fundus and OCT image, including severe cataract of above

grade 2 of nuclear opalescence or cortical cataract according to the Lens Opacities Classification System (LOCS) III [20, 21], posterior subcapsular cataract, or corneal opacities; and (5) other pathologies that could interfere with visual acuity, including glaucoma or intermediate or advanced age-related macular degeneration. This study was approved by the Institutional Review Board of the Pusan National University Yangsan Hospital. Each patient was informed about the risks and benefits of surgery, and written informed consent was obtained. This study adhered to the tenets of the Declaration of Helsinki. Access for data collection was conducted from July 11, 2018 to December 31, 2018. No personal identification information other than the patient's medical record number was included during data collection, and all personal identification information was excluded before data processing.

### Surgical procedure

A 25-gauge standard pars plana vitrectomy was performed using the Constellation system (Alcon Laboratories, Inc., Fort Worth, TX, USA). For all phakic patients, simultaneous phacoemulsification with intraocular lens implantation was performed. The ERM was completely removed using the intraocular forceps (GRIESHABER® Advanced DSP Tips; Alcon Laboratories, Inc., Fort Worth, TX, USA). Additionally, the ILM stained with 0.03% indocyanine green (Diagnogreen; Daiichi-Sankyo Co., Ltd., Tokyo, Japan) was peeled off from all eyes.

### Ocular examination and imaging

All patients underwent comprehensive ophthalmologic examination at baseline and at the 1-, 3-, and 6-month follow-up visits, including best-corrected visual acuity (BCVA) measurement, assessment of metamorphopsia, indirect ophthalmoscopic examination, fundus photography (Canon CR-2 digital non-mydriatic retinal camera, Canon Inc., Tokyo, Japan), OCT, and OCTA. BCVA was measured on a decimal scale using a Snellen chart and then converted to the logarithm of the minimum angle of resolution (logMAR) scale for statistical analysis. The metamorphopsia score (M-score) was assessed using M-chart (Inami Co., Tokyo, Japan) [22]. First, the straight vertical line (0˚) was shown to the patient. If the patient recognized the straight line as curved or irregular, dotted lines with increasing inter-dot spaces on the following pages were presented until they were correctly identified as straight lines. The horizontal M-scores were then calculated. After this step, the M-chart was rotated 90˚ and the vertical scores were checked. The vertical and horizontal M-scores (v and hM-score, respectively) were presented as the degree of the visual angle. The axial length was measured using partial coherence laser interferometry (IOL Master, Carl Zeiss Meditec, Jena, Germany) at baseline. Fundus photography, OCT, and OCTA were also performed on the healthy fellow eyes.

OCT scans were performed using the three-dimensional (3D) macular scan protocol ($6 \times 6$ mm, 256 scan), high-quality 9-mm width B-scan images in 5-line cross mode with 0.25-mm spacing in both the horizontal and vertical directions, and a $6 \times 6$ mm macular cube scan of OCTA. The central subfield macular thickness (CSMT) in a 1-mm area around the fovea was manually measured using a 3D macular scan with built-in software. The ectopic inner foveal layer (EIFL) was characterized by the presence of a continuous hyperreflective or hyporeflective band extending from the inner nuclear and inner plexiform layers over the foveal region evaluated using 3D macular scan and 5-line cross mode [16].

### Assessment of topographic changes in macula: Distance of fovea-disc and fovea-vascular arcades and foveal location

A detailed method for measuring topographic changes in the macula is shown in Fig 1.

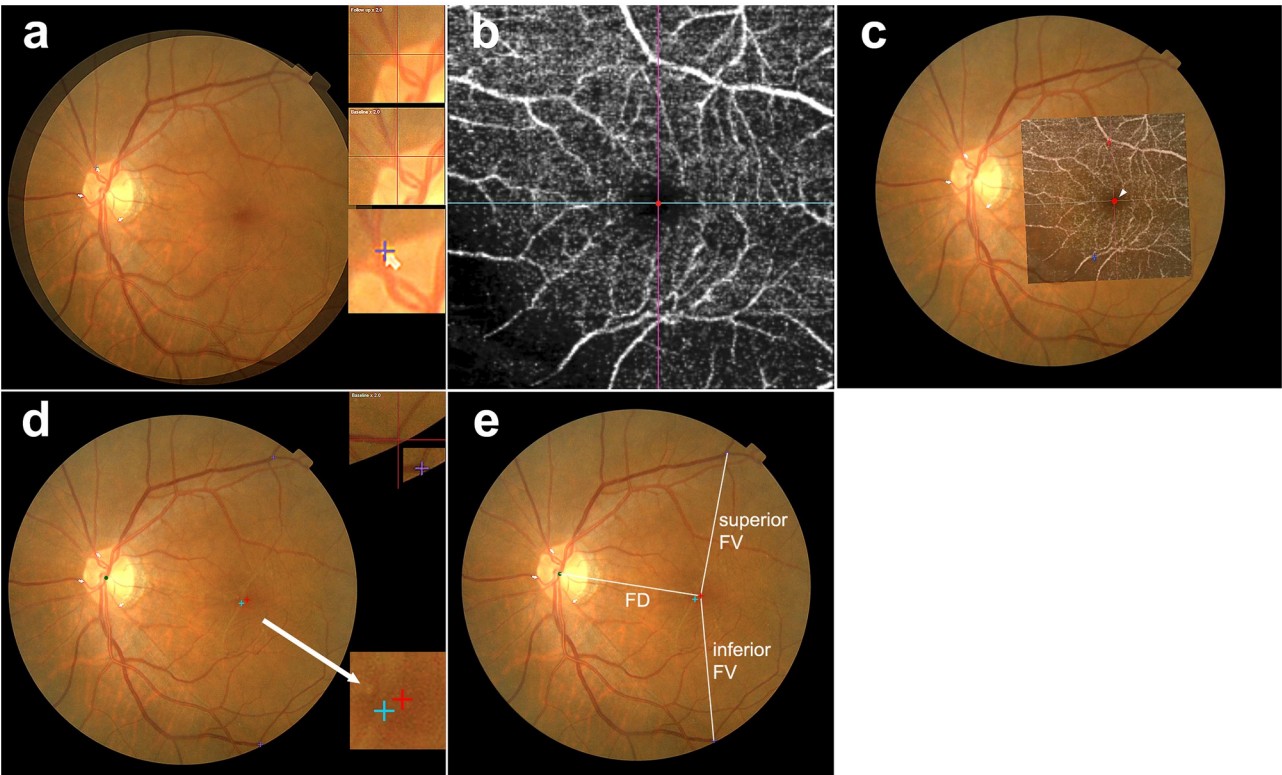

**Fig 1. Methodology for measurement of the macular topographic changes.** (A) Three dots were placed at the junctions of large vessels and the optic disc margin on the fundus image. (B) The center of the fovea was marked at the center of the foveal avascular zone (FAZ) on the optical coherence tomography angiography (OCTA) image. (C) The fundus photograph with dots and the OCTA image with marks were overlapped by matching the retinal vasculature. (D) The baseline and follow-up images were superimposed. (E) The X and Y coordinates of the disc, foveal centers, and bifurcation points of the superior and inferior vascular arcade were identified. Fovea-disc distance and fovea-vascular arcade distance were calculated using these coordinates.

First, baseline and follow-up photographs were superimposed using three dots at the junctions of the large retinal vessels and optic disc margin [23]. Second, the foveal center was identified on the superficial capillary plexus map of the en face OCTA images. If there was no foveal avascular zone (FAZ), the marker was located at the point where the outer nuclear layer was the thickest and the INL was absent or thinnest, using sectional OCT images of the 3D macular cube mode. Using a customized program, the foveal center identified in OCTA images was marked on the corresponding fundus image.

To measure the distances between the fovea and the disc and vascular arcades, the foveal center and bifurcation point of the superior and inferior vascular arcades were marked on the fundus photographs processed as described above. The distances from the fovea to the disc center and vascular arcades were defined as the fovea-disc (FD) and fovea-vascular arcade (FV), respectively. The sum of the superior and inferior FV was defined as vessel-fovea-vessel distance (VFV). The sum of the FD and the superior and inferior FV was defined as the disc-fovea-vessel distance (DFV). After the movement of the fovea was measured in the X and Y coordinates with respect to the disc, to identify foveal dislocation, the X and Y coordinates were recalculated based on the preoperative foveal position. The difference between the affected and fellow eyes was also measured using the unilateral iERM.

## Main outcome measures and statistical analysis

The measured values of parameters used in the study and data on demographic characteristics were organized in S1 File (Data file), and statistical analysis was performed based on this. Statistical analyses were performed using SPSS software (version 22.0; IBM Corp., Armonk, N.Y., USA). Kolmogorov-Smirnov and Shapiro-Wilk analyses were used to test for normality. Continuous variables not normally distributed were expressed as median and interquartile range. Changes in numerical variables, including BCVA, M-score, CSMT, fovea location, FD, FVs, VFV, and DFV, were analyzed using the Wilcoxon signed-rank test or Mann-Whitney U test. Categorical variables were analyzed using the Pearson's chi-square test or Fisher's exact test. The correlation between numerical variables was analyzed using Spearman's correlation test. Binary logistic regression was used to analyze variables associated with visual outcomes. For items with a significance probability of 0.01 or less in univariate binary logistic regression, multiple binary logistic regression was performed using forward stepwise selection (likelihood ratio). The level of statistical significance was set at $P < 0.05$.

## Results

Consecutive 24 eyes of 24 patients (4 males and 20 females) were included in this study. The median age was 64.5 years (interquartile range, 60–71 years). Twenty-one (88%) phakic eyes underwent simultaneous cataract surgery. The EIFL was present in 7 eyes. The baseline characteristics of the patients are summarized in Table 1.

BCVA (logMAR) significantly improved from 0.40 (interquartile range, 0.23–0.50) at baseline to 0.10 (interquartile range, 0.00–0.18) at 6 months follow-up ($P < 0.001$). Fourteen eyes (58.3%) showed visual gain of 3 or more lines. The median baseline vM and hM-score was 0.55˚ (interquartile range, 0.00˚ – 1.60˚) and 0.55˚ (interquartile range, 0.00˚–1.33˚), respectively. At 6 months, vM score significantly improved to 0.10˚ (interquartile range, 0.00˚–0.75˚) ($P = 0.014$), whereas the hM score did not (0.00˚ (interquartile range, 0.00˚–0.90˚), $P = 0.078$). The CSMT significantly decreased from 366.5 μm (interquartile range, 328.0 μm– 471.0 μm)

**Table 1. Baseline characteristics.**

| | |
|---|---|
| Age (years) (median (interquartile range); range) | 64.5 (60–71); 55–78 |
| Sex (male/female) | 4 (17%) / 20 (83%) |
| Lens status (phakic/pseudophakic) | 21 (88%) /3 (12%) |
| Symptom period (22 eyes, months) (median (interquartile range); range) | 12.00 (4.75–18.5); 2–48 |
| Combined phacoemulsification with intraocular lens implantation (eyes) | 21 |
| Refractive error (spherical equivalent, diopter) (median (interquartile range); range) | -0.44 (-1.72–0.88); -5.5 –-2.5 |
| Axial length (mm) (median (interquartile range); range) | 23.55 (23.05–24.20); 22.2–25.9 |
| BCVA (log MAR) (median (interquartile range) | 0.40 (0.23–0.50) |
| M-score (˚) (median (interquartile range); range) | |
| Median vertical M-score | 0.55 (0.00–1.60); 0.0–2.4 |
| Median horizontal M-score | 0.55 (0.00–1.33); 0.0–2.4 |
| CSMT (μm) (median (interquartile range); range) | 366.5 (328.0–471.0); 286–684 |
| Presence of ectopic inner foveal layer (eye) | 7 |

BCVA, best-corrected visual acuity; CSMT, central subfield macular thickness; logMAR, logarithm of the minimum angle of resolution; M-score, metamorphopsia score by M-chart.

to 318.0 μm (interquartile range, 296.3 μm– 363.0 μm) ($P < 0.001$). The EIFL disappeared in 4 (43%) of 7 eyes.

## Topographic changes in the macula

The changes in topographic parameters including FD, FVs, VFV, DFV, and foveal location were assessed by fundus photography, OCT, and OCTA, and their association with CSMT and those of healthy fellow eyes was confirmed. The FD significantly decreased (4.806 mm (interquartile range, 4.425 mm– 5.109 mm) at baseline and 4.595 mm (interquartile range, 4.324 mm– 4.993 mm) at 6 months, $P < 0.001$). Both the superior and inferior FVs significantly increased (3.760 mm (interquartile range, 3.078 mm– 4.496 mm) and 3.480 mm (interquartile range, 3.154 mm– 4.202 mm) at baseline and 4.005 mm (interquartile range, 3.442 mm– 4.646 mm) and 3.740 mm (interquartile range, 3.329 mm– 4.357 mm) at 6-month follow-up, respectively; $P < 0.001$ for both; Fig 2).

These changes led to a significant increase in DFV (11.999 mm (interquartile range, 11.304 mm– 13.247 mm) at baseline and 12.173 mm (interquartile range, 11.567 mm– 13.554 mm) at 6 month follow-u, $P = 0.007$). Baseline vessel-fovea-vessel (VFV) inversely correlated with changes in VFV ($P < 0.001$; Pearson's [r] = -0.672). Eyes with a smaller VFV at baseline still showed a smaller VFV at 6 months ($P < 0.001$, r = 0.984). Table 2 summarizes the postoperative changes in the FD, FV, VFV, and DFV.

Baseline CSMT was positively associated with changes in FD, VFV ($P = 0.017$; r = 0.483, $P = 0.008$; r = 0.531, respectively), and foveal movement ($P < 0.031$). The change in CSMT also correlated with changes in FD ($P = 0.047$; r = 0.410). The foveal center continuously moved towards the optic disc during the follow-up period. The median foveal X coordinates were 153.6 μm, 162.2 μm, and 203.0 μm at 1, 3, and 6 months, respectively, based on the baseline location "0" ($P < 0.001$, respectively, Table 3, Figs 2 and 3).

Compared to healthy fellow eyes, VFV and DFV were significantly lower in the affected eyes throughout the follow-up period ($P < 0.001$ for all). The difference of VFV tended to decrease after surgery (Fig 2, Table 2). The difference in DFV did not show a consistent change. Greater differences in the VFV and DFV were associated with greater foveal movement ($P = 0.010$, r = 0.549 and $P = 0.049$, r = 0.435, respectively).

## Association between topographic factors and visual outcomes

**Change in BCVA.** Eyes with a thicker CSMT presented with worse BCVA at baseline ($P = 0.004$; r = -0.571). Eyes with a greater reduction in CSMT showed greater visual gain ($P = 0.043$; r = 0.416). An increase in the VFV was positively associated with visual gain ($P = 0.049$; r = -0.404). Eyes with visual gain $\geq$ 3 lines showed greater changes in VFV (0.433 mm (interquartile range, 0.294 mm– 0.528 mm) vs 0.249 mm (interquartile range, 0.159 mm– 0.380 mm), $P = 0.036$), compared to others (Table 4).

**Changes in metamorphopsia.** Eyes with thicker CSMT showed greater vM and hM scores at baseline ($P = 0.010$; r = 0.515 and $P = 0.011$; r = 0.511, respectively). The improvement in vM-scores (from 0.55˚ (interquartile range, 0.00˚–1.60˚) to 0.10˚ (interquartile range, 0.00˚–0.75˚), $P = 0.014$) was not associated with changes in topographic parameters, including FD, FVs, VFV, DFV, CSMT, or foveal movement. Eyes with EIFL (n = 7) showed thicker CSMT ($P = 0.024$) and worse vM and hM scores ($P = 0.002$ and 0.009, respectively) at baseline. Those eyes still showed worse vM and hM scores ($P = 0.005$ and 0.005, respectively) at 6-month follow-up despite a greater increase in VFV ($P = 0.02$).

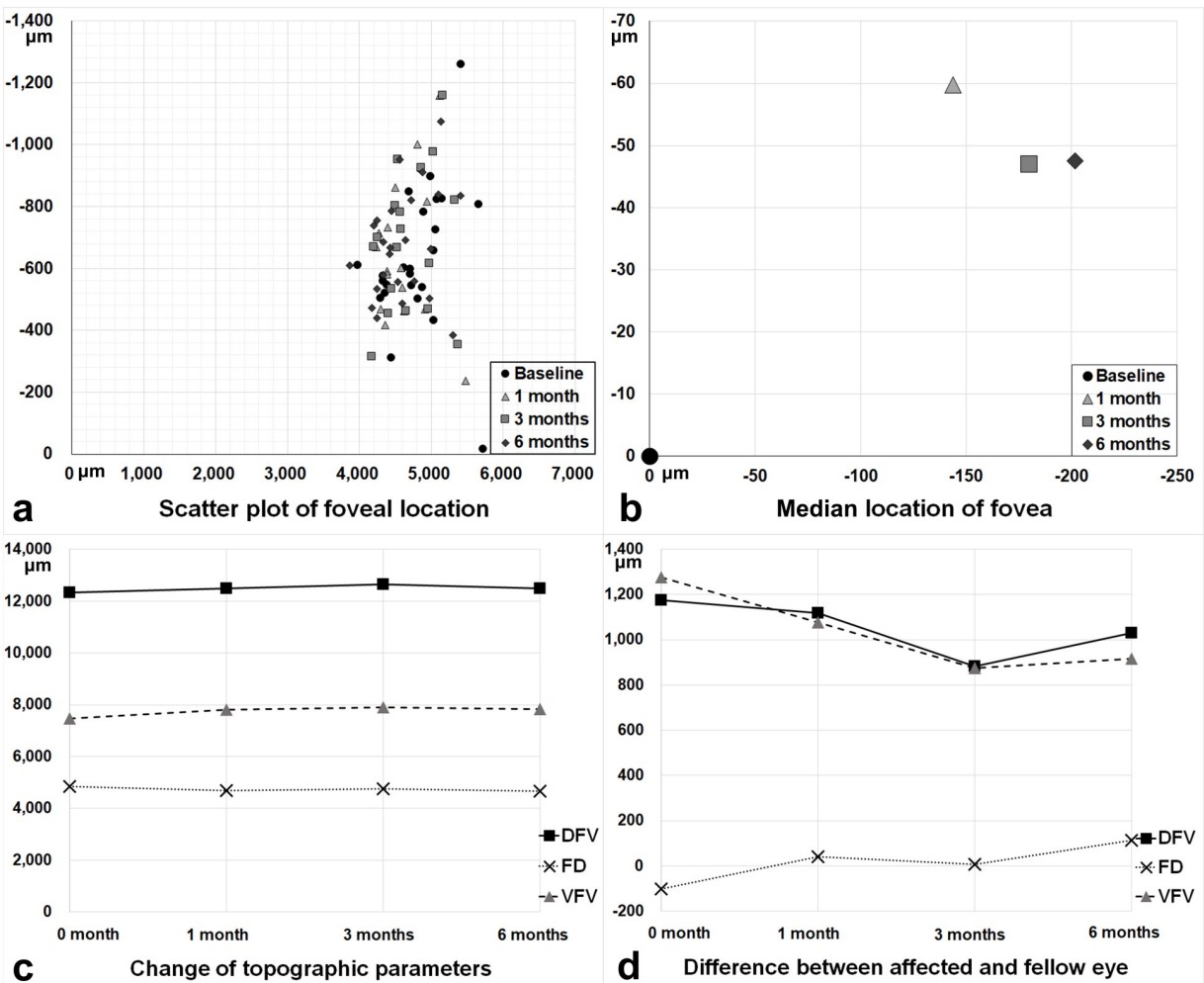

**Fig 2. Change of topographic parameters and foveal location.** (A) Scatter plot of foveal location showed that the position of the fovea tends to move nasally in the horizontal axis and centrally in the vertical axis after surgery. (B) Most of the movement in median foveal location tends to occur within 1 month and the X coordinate continuously moved towards the disc for 6 months. (C) Change of topographic parameters including disc-fovea-vessel distance (DFV), fovea-disc distance (FD), and vessel-fovea-vessel distance (VFV) over 6 months. The distance of VFV increased after surgery, and the distance of FD gradually decreased after surgery, but the amount of change in the distance of FD was relatively small compared to that of VFV. The distance of DFV did not show a consistent change since it was the sum of the FD and VFV distances. (D) Difference in the DFV, FD, and VFV between affected and fellow eyes. Considering the change in the difference of FD, the position of the fovea in the affected eye, which was located temporally compared to the fellow eye before surgery, moved nasally after surgery.

## Factors associated with visual outcomes

Eyes with worse baseline BCVA had worse final visual acuity ($P < 0.015$; r = -0.492) despite greater visual gain ($P < 0.001$; r = -0.735). Overall, changes in BCVA were not associated with sex, age, refractive error, axial length, or lens status. For the factors associated with vision gain $\geq 3$ lines, the univariate logistic analysis presented worse baseline BCVA ($P = 0.017$; odds ratio = $1.065*10^6$), a greater increase in VFV ($P = 0.049$; odds ratio = 1.007), and smaller differences in FD between the affected and fellow eyes at baseline ($P = 0.040$; odds ratio = 0.994) and 6 months ($P = 0.029$; odds ratio = 0.993). Multivariate logistic regression analysis confirmed that a smaller difference in FD between the affected and fellow eyes at 6-month follow-up was associated with visual gain ($P = 0.048$, odds ratio = 0.991).

**Table 2. Changes of topographic parameters in affected eyes and healthy fellow eyes.**

|  | Baseline (24 eyes) | 1 Month (17 eyes) | 3 Months (18 eyes) | 6 Months (24 eyes) | Fellow eye |
|---|---|---|---|---|---|
| FD (mm) (median; interquartile range) | 4.806; 4.425–5.109 | 4.615*; 4.402–4.941 | 4.635*; 4.462–5.040 | 4.595*; 4.324–4.993 | 4.778; 4.562–4.974 |
| Superior FV (mm) (median; interquartile range) | 3.760‡; 3.078–4.496 | 3.706*‡; 3.293–4.763 | 4.039*†‡; 3.512–4.751 | 4.005*‡; 3.442–4.646 | 4.606; 4.129–5.073 |
| Inferior FV (mm) (median; interquartile range) | 3.480‡; 3.154–4.202 | 3.897*‡; 3.389–4.350 | 3.743*‡; 3.356–4.248 | 3.740*‡; 3.329–4.357 | 4.077; 3.768–4.616 |
| DFV (mm) (median; interquartile range) | 11.999‡; 11.304–13.247 | 12.105*‡; 11.466–13.664 | 12.309*†‡; 11.653–13.935 | 12.173*‡; 11.567–13.554 | 13.597; 12.519–14.077 |
| VFV (mm) (median; interquartile range) | 7.114‡ 6.563–8.302 | 7.208*‡; 6.908–8.725 | 7.396*†‡; 7.144–8.830 | 7.369*‡; 7.046–8.838 | 8.739; 8.064–9.273 |

FD, fovea-disc distance; FV, fovea-vessel distance; DFV, disc-fovea-vessel distance; VFV, vessel-fovea-vessel distance.

*$P < 0.05$ compared with baseline,

†$P < 0.05$ compared with previous follow up,

‡$P < 0.05$ compared with healthy fellow eye.

## Discussion

This study highlights significant changes in macular topography after ERM removal. The fovea moved horizontally towards the optic disc, and the distance between the superior and inferior vascular arcades increased. Eyes with thicker CSMT showed greater changes in FD, VFV, and foveal movement. BCVA and vertical metamorphopsia significantly improved. Visual gain was associated with an increased distance between the vascular arcades and decreased macular thickness. However, no topographic factors were relevant to the improvement in vertical metamorphopsia. This study investigated the association between macular topographic changes and visual acuity via multimodal imaging comprising fundus photographs, OCT, and OCTA.

The ERM is a tissue composed of myofibroblasts and an extracellular matrix in front of the macula. Macular deformity from gradual membrane contraction induces visual impairment and metamorphopsia [8, 9, 16, 24]. The structural changes can be observed as follows: decreased distance between the fovea, disc, and vessels; retinal blood vessel tortuosity; ectopic foveal location; and thickened and wrinkled macular tissue [16, 24–26]. Such anatomical changes worsen over time, resulting in photoreceptor damage detected as disrupted ellipsoid zones (EZ) on OCT, and cause irreversible visual loss [13–15]. We included eyes with intact

**Table 3. Changes in foveal location after surgery.**

|  | Baseline (24 eyes) | 1 Month (18 eyes) | 3 Months (18 eyes) | 6 Months (24 eyes) |
|---|---|---|---|---|
| X coordinate (median; interquartile range) | 0 0 | -153.6*; -228.0 – -72.1 | -162.2*; -268.9 –-123.5 | -203.0*†; -279.6 – -115.7 |
| Y coordinate (median; interquartile range) | 0 0 | -54.6*; -108.7–9.8 | -54.0; -151.8–54.2 | -37.3; -174.0–71.4 |

The X and Y coordinates were calculated based on the preoperative foveal location. The left eye was used as the reference eye, and the superior and lateral sides were considered as positive directions.

*$P < 0.05$ compared with baseline,

†$P < 0.05$ compared with previous follow up

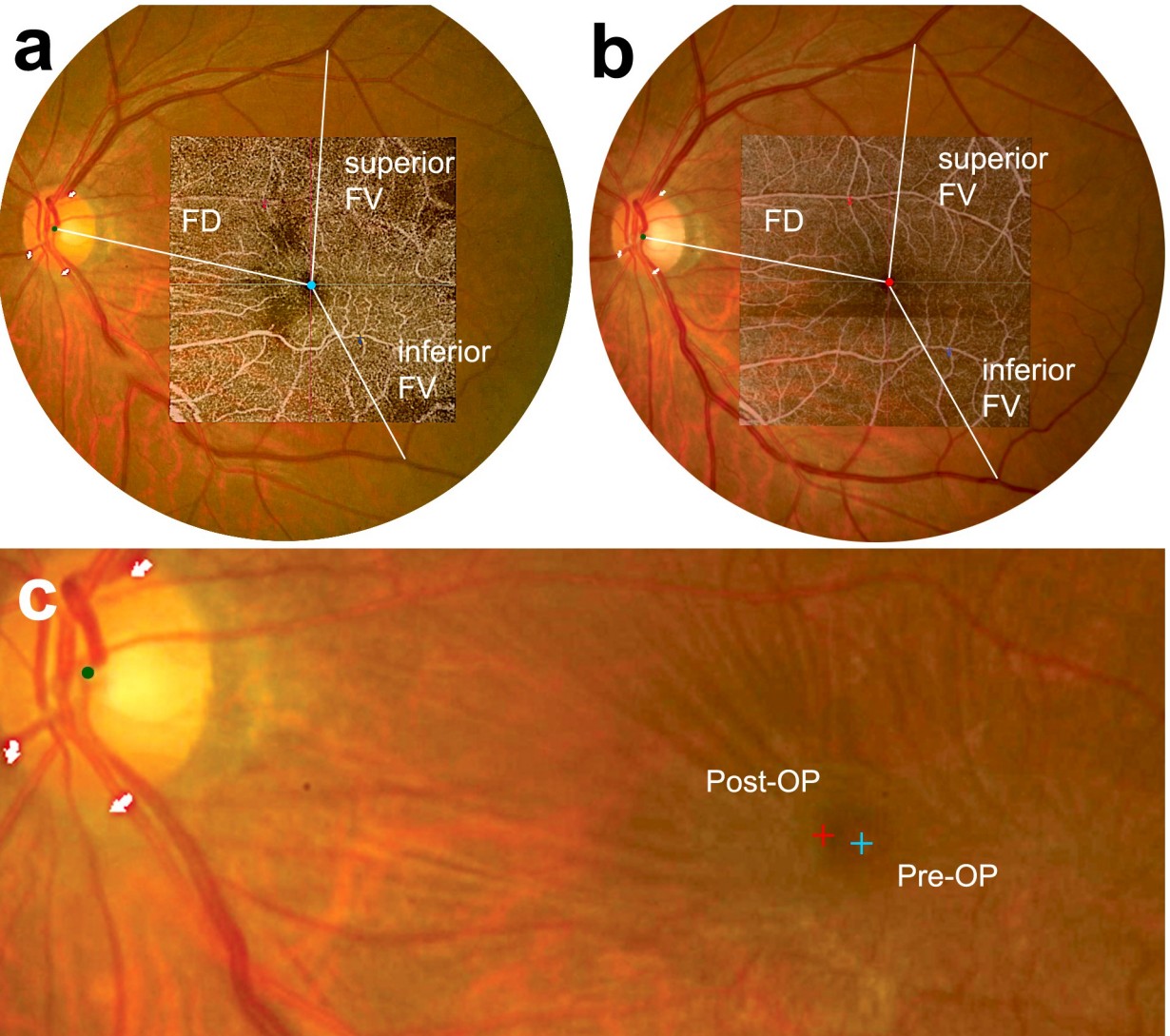

**Fig 3. Representative case showing topographic change after epiretinal membrane (ERM) removal.** (A) On the baseline fundus photograph, after the point of vascular origin (green dot) on the disc, representing the center of the disc, and the center of the fovea (blue dot), defined as the center of the foveal avascular zone on optical coherence tomography angiography, were marked and the distance of fovea-disc (FD), superior fovea-vascular distance (FV) and inferior FV were calculated. (B) On the follow-up fundus photograph, the same process was applied as above. The center of the fovea was marked by a red dot. The superior and inferior FVs increased with release of membrane contraction. The FD decreased after surgery. Three points of vessels crossing the disc rim (white arrows) were marked to superimpose the baseline and follow-up fundus photographs. (C) The preoperative and postoperative foveal location were marked on the baseline fundus photograph.

EZ on OCT because surgical treatment is commonly recommended before severe photoreceptor damage develops.

Early visual loss can develop by tangential contraction of the macula without thickening [24–26]. This finding indicates that tangential contraction is the main cause of disease progression and topographic assessment of the macula is important in eyes with iERM. Several studies have assessed the topographic changes of the macula using post-surgery fundus photography [25, 27–32]. The tangential changes of the macula were assessed using various methodologies as follows: the nasal area and angle between the inferior and superior vascular arcades [28], the vessel segment length and movement in a particular region [29], the distance between vascular

**Table 4. Comparison between eyes with and without visual gain of 3 lines or more.**

| | Eyes with visual gain | Eyes without visual gain | *P*-value |
|---|---|---|---|
| Eyes (number) | 14 | 10 | |
| Median BCVA (log MAR) (median; interquartile range) | | | |
| Baseline | 0.40; 0.40–0.55 | 0.20; 0.20–0.40 | .002 |
| Month 6 | 0.10; 0.00–0.10 | 0.10; 0.00–0.30 | .371 |
| CSMT (µm) (median; interquartile range) | | | |
| Baseline | 382.5; 351.3–491.3 | 362.5; 304.3–429.0 | .212 |
| Month 6 | 318.0; 303.0–398.3 | 321.5; 283.8–348.3 | .585 |
| Δ FD both eye (mm) (median; interquartile range) | | | |
| Baseline | -0.274*; -0.649 –-0.018 | 0.083*; 0.154–0.302 | 0.004 |
| Month 6 | -0.019; -0.384–0.207 | 0.398*; 0.267–0.498 | 0.006 |
| Δ VFV both eye (mm (median; interquartile range) | | | |
| Baseline | 1.394*; 0.744–1.548 | 1.158*; 0.874–2.148 | 0.808 |
| Month 6 | 0.989*; 0.251–1.232 | 0.902*; 0.619–1.763 | 0.972 |
| Δ DFV both eye (mm) (median; interquartile range) | | | |
| Baseline | 0.881*; 0.326–1.570 | 1.338*; 1.071–2.310 | 0.310 |
| Month 6 | 0.692*; 0.028–1.444 | 1.274*; 1.060–2.068 | 0.193 |

BCVA, best corrected visual acuity; CSMT, central subfield macular thickness; Δ DFV, difference of disc-fovea-vessel; Δ FD, difference of fovea-disc distance; log MAR, logarithm of minimum angle of resolution; M-score, metamorphopsia score by the M-chart; Δ difference of VFV, vessel-fovea-vessel distance.

*P-value of difference between both eyes was < 0.05.

arcades across the fovea [25], the distance from the fovea to the small retinal vessels around the fovea [30], and the distances between the intersections of retinal vessels around the fovea [31, 32]. Most of these studies observed that significant changes in their topographic parameters were associated with the OCT parameter of macular thickness. However, many of them did not confirm the association between the changes in topographic parameters and BCVA following surgery [28–32]. The conflicting outcomes of the association between changes in macular topography and visual gain would result from the different methodologies, particularly the extent of the macula covered by the parameters. In previous studies, although significant changes were observed, the angle and nasal area between the vascular arcades and the distance changes of retinal vessels in specific regions or around fovea could not reflect the topographical changes in the entire macula, and the association between parameters and visual improvement would not be found [28, 29]. In our observation, VFV (distance from foveola to the superior and inferior vascular arcade) was a topographic parameter associated with visual improvement. Because the decreased distance between the foveola and the optic disc minimized the increase in VFV, no association between the change in DFV (a parameter for overall macular topography) and visual acuity was found.

Regarding metamorphopsia, topographic parameters reflecting the area near the fovea, including changes in macular topography, are valuable. Increased distance between the vertical and horizontal lines across the fovea (which did not reach the vascular arcade) has reportedly been associated with improvement in metamorphopsia [31, 32]. Additionally, Ichikawa et al.

reported that INL thickness on OCT correlated with topographic factors and metamorphopsia [33]. However, none of the topographic factors showed an association with improvement in metamorphopsia in the present study.

Longitudinal OCT and OCTA revealed changes in macular thickness and foveal location. A thicker baseline macula was associated with greater changes in FD and VFV, and more horizontal movement of the fovea towards the optic disc. Foveal movement was not associated with visual outcomes. Tung et al. also demonstrated that the nasal shift of the fovea observed on infrared fundus photography had no correlation with visual improvement after ERM removal [34], which is similar to our findings. Difference of foveal movement (162.42 μm in 24 months vs 203.0 μm in 6 months) might be derived from the difference in devices used to detect the fovea center (infrared fundus photography vs OCTA) and patients' characteristics. Detection of the foveola in eyes with ERMs was more accurate on OCTA than on fundus photography, due to the high-resolution in-depth scan images. Horizontal movement of the fovea may result from the pathological characteristics of ERM. When the ERM contracts the macula centripetally, the retinal layers become distorted and thickened within the small, contracted macula [23–25, 27]. At this stage, the fovea has no choice but to move temporally because the optic disc acts like a wall to block nasal movement. Therefore, the fovea relocates nasally owing to the centrifugal movement of the macular structure after ERM removal (elimination of centripetal traction). Moreover, ILM peeling may have played an additional role in such foveal movement. Several studies of macular hole and diabetic macular edema have shown foveal movement towards the optic disc after ILM peeling. This was believed to be caused by contraction of the nerve fiber layer [31, 34, 35]. Neither previous studies nor our present study showed an association between foveal movement and metamorphopsia. Regarding the above-mentioned topographic factors, metamorphopsia may be related more to the surrounding structural changes around the fovea than foveal movement alone. This reasoning about metamorphopsia have also been suggested in the context of macular hole [35–37].

This study had several limitations, including its retrospective nature and small sample size. The effect of ILM removal on topographic changes of macula was inconclusive in this study, because ILM was peeled in all eyes. Although eyes with visually affecting cataract (more than grade 2 nuclear opalescence or cortical cataract) were excluded, mild cataract could be a potential confounding factor in assessing visual acuity. Nevertheless, based on previous studies for postoperative cataract changes, nuclear opalescence and cortical cataract of grade 2 or lower had little effect on visual loss [38, 39]. Regarding these, minimal cataract changes in phakic eyes had little effect on the postoperative visual acuity during a 6-month follow-up. Furthermore, detailed tomographic factors such as the INL, central foveal bouquet, and others were not investigated. These factors can also influence visual acuity and metamorphopsia. We may also obtain different outcomes if eyes with iERM were classified and distinguished based on severity. Further studies are required to analyze these factors. Nevertheless, we found that the topographic assessment of the macula using multimodal imaging provided useful metrics for predicting surgical outcomes in iERM.

## Conclusions

In our study, the macular topography significantly changed after iERM removal. The fovea moved nasally, and the distance between the superior and inferior vascular arcades increased, indicating the release of tangential contraction on the macula. Among the topographic parameters, changes in VFV were associated with improvements in BCVA. However, none of these factors was associated with changes in metamorphopsia.

## Supporting information

**S1 File. Data file used in this study.**
(XLSX)

## Author Contributions

**Conceptualization:** Seung Min Lee, Iksoo Byon.

**Data curation:** Seung Min Lee.

**Formal analysis:** Seung Min Lee, Iksoo Byon.

**Investigation:** Seung Min Lee, Iksoo Byon.

**Methodology:** Seung Min Lee, Iksoo Byon.

**Supervision:** Iksoo Byon.

**Validation:** Sung Who Park.

**Visualization:** Seung Min Lee.

**Writing – original draft:** Seung Min Lee.

**Writing – review & editing:** Sung Who Park, Iksoo Byon.

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
