## [Decision Letter · Decision Letter 0]

19 Aug 2024

PONE-D-24-31532Topographic Changes in Macula and Its Association with Visual Outcomes in Idiopathic Epiretinal Membrane SurgeryPLOS ONE

Dear Dr. Byon,

Thank you for submitting your manuscript to PLOS ONE. After careful consideration, we feel that it has merit but does not fully meet PLOS ONE’s publication criteria as it currently stands. Therefore, we invite you to submit a revised version of the manuscript that addresses the points raised during the review process. both reviewers found the paper worthwhile and made some suggestions for improvement. We look forward to the revised version 

We look forward to receiving your revised manuscript.

Kind regards,

Demetrios G. Vavvas

Academic Editor

PLOS ONE

Journal Requirements:

2. In the online submission form, you indicated that [The data underlying the results presented in the study are available by contacting the author.]. 

Reviewers' comments:

Reviewer's Responses to Questions

**Comments to the Author**

1. Is the manuscript technically sound, and do the data support the conclusions?

Reviewer #1: Yes

Reviewer #2: Yes

2. Has the statistical analysis been performed appropriately and rigorously? 

Reviewer #1: Yes

Reviewer #2: Yes

3. Have the authors made all data underlying the findings in their manuscript fully available?

Reviewer #1: Yes

Reviewer #2: Yes

4. Is the manuscript presented in an intelligible fashion and written in standard English?

Reviewer #1: Yes

Reviewer #2: No

5. Review Comments to the Author

Reviewer #1: Notes:

1.Change: “The idiopathic epiretinal membrane (iERM) commonly appears in older people as

glistening amorphous membranes on the macula” to “Idiopathic Epiretinal membrane commonly appears in the older population as a glistening amorphous membrane on the macula”

2. Would also include in the references about previous studies the following study: Vingopoulos F, Koulouri I, Miller JB, Vavvas DG. Anatomical and Functional Recovery Kinetics After Epiretinal Membrane Removal. Clin Ophthalmol. 2021;15:175-181

https://doi.org/10.2147/OPTH.S264948

Reviewer #2: Peer Review Report (also attached)

Summary of the Manuscript:

The manuscript explores the macular topographic changes and their association with visual

acuity changes and metamorphopsia in patients with idiopathic epiretinal membranes (iERM).

This was executed through a retrospective chart review of 24 eligible consecutive patients who

underwent iERM peel and were evaluated over the course of 6 months post-operatively. The

authors aim to determine whether multimodal imaging of fundus photographs, optic coherence

tomography (OCT) and OCT-angiography (OCT-A) may be of prognostic value regarding visual

acuity and improvement of metamorphopsia.

Overall Evaluation:

This study provides valuable information on how changes in anatomical findings of the macula

can help us in having an idea about what to expect in terms of visual prognosis after ERM peel

surgery. The overall methodology is a significant strength of the study, but there are several

areas where the manuscript could be improved, as mentioned below:

Comments:

1. Introduction:

o The manuscript’s introduction successfully provided background information on

iERM as it elaborated on associated exam findings, visual outcomes and

symptoms. However, it needs to better explain that the aim was to compare the

macular topography changes, visual acuity and metamorphopsia pre-operatively

versus post-operatively as part of establishing prognosis (as depicted in the title

and conclusion).

2. Results:

o Statistics in tables 1 through 4 have been reported as mean +- SD; range.

However, non-parametric tests were used for further data analysis, meaning that

the data obtained was not normally distributed. A better way to present central

tendency and measures of dispersion in that case is median (Q1, Q3) or median

[min-max].

o In table 1, baseline BCVA is reported as a mean of 0.65 +- 0.25. In the text below

it, it is said that baseline BCVA significantly improved from 0.42 +- 0.22 at

baseline to 0.12 +- 0.15 at 6 months. Which value is the correct baseline BCVA?

3. Discussion:

o The discussion does highlight the important points to be covered, but more

concrete findings and examples should be given when the authors mention that

many studies looked at topographic changes in the macula after surgery but did

not correlate them with BCVA due to conflicting findings regarding the

methodology.

4. Clarity and Structure:

o Regarding the methodology behind the calculation of the fovea-disc and foveaarcades distances, I would suggest to briefly explain the steps done in order to do

so and then mention that further details are shown in Figure 1 instead of the other

way around, as the legend gives a great stepwise explanation on how this was

done.

o Figures 1 and 3 are a little blurry. Higher quality figures would make them easier

to navigate, particularly for figure 1 as the edges of the optic disc are important

landmarks to the methodology behind the study.

Minor Comments:

o Introduction, page 1: “the gradual distortion of the retinal structure” can be

replaced with “the gradual distortion of the retinal layers.”

o Introduction, page 1: the words “most” and “have” in the sentence “However,

most studies have not used multimodal imaging” could be replaced with the

words “previous” and “did”, respectively.

o Introduction, page 1: the sentence “This methodological weakness may contribute

to the current discrepancy in the prognostic values of some factors” is ambiguous.

It would be a good idea to rephrase it to explain the fact that previously studied

methodologies failed to account for confounding variables with ERM-affected

eyes that might affect prognosis of surgery. The example that follows that

sentence illustrates the idea really well.

o Materials and methods, study participants, page 3: add the words “with an” to the

sentence “Eyes presenting (with an) idiopathic epiretinal membrane that typically

covered the fovea”

o Materials and methods, study participants, page 3: add a dash between the words

“age” and “related” to describe age-related macular degeneration.

o Material and methods, Ocular examination and Imaging, page 4: add the words

“measurement” and “assessment of” for better clarity: “All patients underwent

comprehensive ophthalmologic examination at baseline and the 1-, 3- and 6-

month follow-up visits, including best-corrected visual acuity (BCVA)

(measurement), (assessment of) metamorphopsia.”

o Material and methods, Ocular examination and Imaging, page 5: add the word

“also” in the sentence “Fundus photography, OCT, and OCTA were (also)

performed on the healthy fellow eyes”.

o Material and methods, main outcomes measures and statistical analysis, page 6:

replace “for normality tests” with “to test for normality” in “KolmogorovSmirnov and Shapiro-Wilk analyses were used for normality tests.”

o Material and methods, main outcomes measures and statistical analysis, page 7:

replace “the factors” with “variables” in “Binary logistic regression was used to

analyze the factors associated with visual outcomes.”

o Results, page 8: add the word “present” in the sentence “The EIFL was (present)

in 7 eyes.”

o Table 1: it is visually better to present sex and lens status as proportions using

percentages instead of solely reporting absolute numbers.

o Table 1: It should be mentioned what the measures used for the symptom period

are i.e. (mean +-SD, range).

o Table 1: A typo in the word "months" (misspelled as "month") next to “symptom

period” should be corrected.

o Results section, under “Topographic changes in the Macula”, first sentence, page

9: the terms “the changes in parameters” and “their associations” are ambiguous.

It would be helpful to the reader to define what those entail.

o Results, page 9, in Fig. 2 legend: “the X coordinate moved to the disc after 6

months” could be replaced with “the X coordinate moved towards the disc after 6

months.”

o Figure 2 legend, page 9: I suggest that the last 2 sentences of C and D be included

as part of the results section of the manuscript and not as part of the legend, as

they describe findings. I would also just describe what the graphs are about in (A)

and (B) and explain the findings in the manuscript.

o Results, page 10: could replace the word “to” with “towards” in the sentence: “the

fovea continuously moved to the optic disc during the follow-up period.”

o Results, page 11, figure 3 legend: remove the dash in “blue-cross” and “redcross”.

o Results, page 11, figure 3 legend: remove the dash in “blue-cross” and “redcross”.

o Results, page 11, figure 3 legend: could remove the last sentence as this was

already mentioned in the results and will be reiterated in the discussion.

o Results, bottom of page 11, under Association between topographic factors and

visual outcomes: add the word “with” in the sentence: “eyes with a thicker CSMT

presented (with) worse BCVA at baseline.”

o Results, bottom of pages 11 and 12, under Association between topographic

factors and visual outcomes: “change in BCVA” and “change in

metamorphopsia” should be subtitles for better clarity.

o Results, page 13: replace “such” with” “those” in “Such eyes still showed worse

vM and hM scores”

o Discussion, second sentence, page 14: “the fovea moved horizontally to the optic

disc” could be replaced with “the fovea moved horizontally towards the optic

disc”, implying that it moved nasally, as the results described.

o Discussion, page 14, last sentence of first paragraph can be rewritten as follows:

“This study investigated the association between macular topographic changes

and visual acuity via multimodal imaging comprising fundus photographs, OCT,

and OCTA.”

o Discussion, page 15: A typo in the word "INL" (misspelled as "ILN") in the

sentence “ILN thickness on OCT” should be corrected.

o Discussion, page 16: A typo in the word "disc" (misspelled as "disk") in the

sentence “more horizontal movement of the fovea to the optic disk” should be

corrected.

o Discussion, page 16: “more horizontal movement to the optic disc” could be

replaced with “more horizontal movement towards the optic disc.”

o Discussion, page 16: “foveal movement to the optic disc” could be replaced with

“foveal movement towards the optic disc.”

o Discussion, page 16: add the terms “in the context of” for better clarity in the

sentence: “This reasoning about metamorphopsia have also been suggested (in the

context of) macular hole.”

o Discussion, page 17: can rephrase “Different outcomes can be achieved if eyes

with iERM are classified and distinguished based on severity” to “We may also

obtain different outcomes if eyes with iERM were classified and distinguished

based on severity” for better flow of the paragraph.

o In figure 1 (E), the text is not very clear.

o In figure 1, I would make the red and blue crosses more apparent (i.e. more bold

and sharper borders). I would also consider changing the color of the red cross to

something with higher contrast as red blends in with the color fundus photo’s

background.

o In figure 3, I would suggest replacing the red dot at the optic disc with another

color such a black or green as red might be mistaken as part of retinal blood

vessels.

o In figure 3, make the blue cross more apparent in (A) and the red cross more

apparent in (B). Again, in (B) and (C ), I would also suggest changing the color of

the red cross for the same reason discussed above for figure 1.

6. PLOS authors have the option to publish the peer review history of their article (what does this mean?). If published, this will include your full peer review and any attached files.

Reviewer #1: No

Reviewer #2: No

---

## [Author Response · Author response to Decision Letter 0]

7 Sep 2024

I. Journal Requirements:

Response: We confirmed that our manuscript complies with PLOS ONE's style requirements.

2. In the online submission form, you indicated that [The data underlying the results presented in the study are available by contacting the author.]. 

Response: The data has been uploaded as a supplementary information file.

Response: It was already described in the first uploaded manuscript. IRB number, etc. are written in the additional information section of the editorial manager.

Response: There was one reference added according to the reviewer's request, and there were no other special modifications.

II. Point-by-point response to the reviewer’s comments

Dear Editor & Reviewers,

We appreciate all suggestions and corrections. All authors have carefully reviewed these comments. Our detailed responses to comments are addressed below.

Review Comments to the Author

Reviewer #1: Notes:

1.Change: “The idiopathic epiretinal membrane (iERM) commonly appears in older people as glistening amorphous membranes on the macula” to “Idiopathic Epiretinal membrane commonly appears in the older population as a glistening amorphous membrane on the macula”

Response 1: As the reviewer’s comments, the corresponding sentence has been corrected (First sentence of introduction).

2. Would also include in the references about previous studies the following study: Vingopoulos F, Koulouri I, Miller JB, Vavvas DG. Anatomical and Functional Recovery Kinetics After Epiretinal Membrane Removal. Clin Ophthalmol. 2021;15:175-181

https://doi.org/10.2147/OPTH.S264948

Response 2: The recommended reference was added according to the reviewer’s comment (Line 9 of the second paragraph in the introduction).

Reviewer #2: Peer Review Report (also attached)

Summary of the Manuscript:

The manuscript explores the macular topographic changes and their association with visual acuity changes and metamorphopsia in patients with idiopathic epiretinal membranes (iERM). This was executed through a retrospective chart review of 24 eligible consecutive patients who underwent iERM peel and were evaluated over the course of 6 months post-operatively. The authors aim to determine whether multimodal imaging of fundus photographs, optic coherence tomography (OCT) and OCT-angiography (OCT-A) may be of prognostic value regarding visual acuity and improvement of metamorphopsia.

Overall Evaluation:

This study provides valuable information on how changes in anatomical findings of the macula can help us in having an idea about what to expect in terms of visual prognosis after ERM peel surgery. The overall methodology is a significant strength of the study, but there are several areas where the manuscript could be improved, as mentioned below:

Comments:

1. Introduction:

o The manuscript’s introduction successfully provided background information on ERM as it elaborated on associated exam findings, visual outcomes and symptoms. However, it needs to better explain that the aim was to compare the macular topography changes, visual acuity and metamorphopsia pre-operatively versus post-operatively as part of establishing prognosis (as depicted in the title and conclusion).

Response: As comments of reviewer 1 and 2, we revised the introduction.

2. Results:

o Statistics in tables 1 through 4 have been reported as mean +- SD; range. However, non-parametric tests were used for further data analysis, meaning that the data obtained was not normally distributed. A better way to present central tendency and measures of dispersion in that case is median (Q1, Q3) or median [min-max].

Response: In agreement with the reviewer’s opinion, the data in the results and tables 1 through 4 were changed to median and interquartile range. 

In statistical analysis, “Continuous variables not normally distributed were expressed as median and interquartile range.” was added.

o In table 1, baseline BCVA is reported as a mean of 0.65 +- 0.25. In the text below it, it is said that baseline BCVA significantly improved from 0.42 +- 0.22 at baseline to 0.12 +- 0.15 at 6 months. Which value is the correct baseline BCVA?

Response: The correct figure was 0.42 +- 0.22 which had been presented in the text. There appears to have been a typo. Expressed as median and interquartile range, it was changed to 0.40 (interquartile range, 0.23 – 0.50) (Results, Table 1).

3. Discussion:

o The discussion does highlight the important points to be covered, but more concrete findings and examples should be given when the authors mention that many studies looked at topographic changes in the macula after surgery but did not correlate them with BCVA due to conflicting findings regarding the methodology.

Response: In the third paragraph of the Discussion, various studies that investigated tangential contraction and the characteristic research methods for each study are described as follows. 

“The tangential changes of the macula were assessed using various methodologies as follows: the nasal area and angle between the inferior and superior vascular arcades, the vessel segment length and movement in a particular region, the distance between vascular arcades across the fovea, the distance from the fovea to the small retinal vessels around the fovea, and the distances between the intersections of retinal vessels around the fovea“. 

In addition, it was stated that most of these studies without multimodal method failed to determine the relationship between visual acuity and topological change (“However, many of them did not confirm the association between the changes in topographic parameters and BCVA following surgery”). 

In the sentence “Most studies observed that significant changes in their topographic parameters were associated with the OCT parameter of macular thickness.”, the expression “Most studies” had been changed to “Most of these studies” to emphasize the connection with the previous sentence. 

4. Clarity and Structure:

o Regarding the methodology behind the calculation of the fovea-disc and fovea arcades distances, I would suggest to briefly explain the steps done in order to do so and then mention that further details are shown in Figure 1 instead of the other way around, as the legend gives a great stepwise explanation on how this was done.

Response: In the methods, the description of techniques used to measure variables had been changed briefly as follows: 

from “First, we placed three dots at the junctions of the large retinal vessels and optic disc margin to serve as landmarks. We then superimposed the baseline and corresponding follow-up fundus photographs to confirm their concurrence and minimize the mismatch of features between the two photographs [23]. Second, the foveal center was identified on the superficial capillary plexus map of the en face OCTA images. If there was no foveal avascular zone (FAZ), the marker was located at the point where the outer nuclear layer was the thickest and the inner nuclear layer was absent or thinnest, using sectional OCT images of the 3D macular cube mode. The OCTA images were transformed and overlapped by matching the vasculature using a customized program, and the foveal center was marked on the corresponding fundus images.” to “First, baseline and follow-up photographs were superimposed using three dots at the junctions of the large retinal vessels and optic disc margin [23]. Second, the foveal center was identified on the superficial capillary plexus map of the en face OCTA images. If there was no foveal avascular zone (FAZ), the marker was located at the point where the outer nuclear layer was the thickest and the INL was absent or thinnest, using sectional OCT images of the 3D macular cube mode. Using a customized program, the foveal center identified in OCTA images was marked on the corresponding fundus images.” (Materials and methods, Distance of Fovea-Disc and Fovea-Vascular Arcades and Foveal Location, 1st paragraph, 2nd sentence).

o Figures 1 and 3 are a little blurry. Higher quality figures would make them easier to navigate, particularly for figure 1 as the edges of the optic disc are important landmarks to the methodology behind the study.

Response: As the reviewer mentioned, the positioning of the landmarks on the edges of the optic disc was most important in this study and could have the greatest impact on reproducibility. Fig 1 presents the methodology using multimodal images as it was actually carried out. The reason Fig 1a is blurred is because each image has been changed to transparent with low opacity to overlap the two images. Small, illegible letters in Fig 1e have been removed. The blurry parts in Figure 3C have been somewhat ameliorated. However, since the image has been greatly enlarged, there are limitations in improving the blurry parts due to problems with the original image quality. Due to the large size of the figure file, there was a slight deterioration in image quality after image processing using PACEV provided by PLOS ONE, but considering the nature of the electronic journal, accurate structures and markers can be confirmed by viewing the uploaded original image rather than printed size image. (Figure 1 and 3)

Minor Comments:

o Introduction, page 1: “the gradual distortion of the retinal structure” can be

replaced with “the gradual distortion of the retinal layers.”

Response: The corresponding sentence has been corrected according to the reviewer’s comment (Introduction, 1st paragraph, 6th line)

o Introduction, page 1: the words “most” and “have” in the sentence “However, most studies have not used multimodal imaging” could be replaced with the words “previous” and “did”, respectively.

Response: The corresponding sentence has been corrected according to the reviewer’s comment (Introduction, 3rd paragraph, 1st line)

o Introduction, page 1: the sentence “This methodological weakness may contribute to the current discrepancy in the prognostic values of some factors” is ambiguous. It would be a good idea to rephrase it to explain the fact that previously studied methodologies failed to account for confounding variables with ERM-affected eyes that might affect prognosis of surgery. The example that follows that sentence illustrates the idea really well.

Response: The corresponding sentence has been corrected according to the reviewer’s comment: From “This methodological weakness may contribute to the current discrepancy in the prognostic values of some factors (e.g., various factors influencing each other in eyes with iERM, which can be confounding factors in predicting prognosis).” to “Various factors influence each other in eyes with iERM, which can be confounding factors in predicting prognosis. Previously studied methodologies were hard to figure out confounding variables that might affect prognosis of surgery.”. (Introduction, 3rd paragraph, 2nd sentence).

o Materials and methods, study participants, page 3: add the words “with an” to the sentence “Eyes presenting (with an) idiopathic epiretinal membrane that typically covered the fovea”

Response: The corresponding sentence has been corrected according to the reviewer’s comment (Materials and methods, study participants, 2nd paragraph, 2nd sentence). 

o Materials and methods, study participants, page 3: add a dash between the words “age” and “related” to describe age-related macular degeneration.

Response: The corresponding sentence has been corrected according to the reviewer’s comment (Materials and methods, study participants, 2nd paragraph, 13th line). 

o Material and methods, Ocular examination and Imaging, page 4: add the words “measurement” and “assessment of” for better clarity: “All patients underwent comprehensive ophthalmologic examination at baseline and the 1-, 3- and 6- month follow-up visits, including best-corrected visual acuity (BCVA) (measurement), (assessment of) metamorphopsia.”

Response: The corresponding sentence has been corrected according to the reviewer’s comment (Materials and methods, Ocular Examination and Imaging, 1st paragraph, 1st sentence).

o Material and methods, Ocular examination and Imaging, page 5: add the word “also” in the sentence “Fundus photography, OCT, and OCTA were (also) performed on the healthy fellow eyes”.

Response: The corresponding sentence has been corrected according to the reviewer’s comment. (Materials and methods, Ocular Examination and Imaging, 1st paragraph, the last sentence).

o Material and methods, main outcomes measures and statistical analysis, page 6: replace “for normality tests” with “to test for normality” in “KolmogorovSmirnov and Shapiro-Wilk analyses were used for normality tests.”

Response: The corresponding sentence has been corrected according to the reviewer’s comment (Materials and methods, main outcome measures and statistical analysis, 1st paragraph, 2nd sentence).

o Material and methods, main outcomes measures and statistical analysis, page 7: replace “the factors” with “variables” in “Binary logistic regression was used to analyze the factors associated with visual outcomes.”

Response: The corresponding sentence has been corrected according to the reviewer’s comment (Materials and methods, main outcome measures and statistical analysis, 1st paragraph, 8th line).

o Results, page 8: add the word “present” in the sentence “The EIFL was (present) in 7 eyes.”

Response: The corresponding sentence has been corrected according to the reviewer’s comment (Results, 1st paragraph, 3rd sentence).

o Table 1: it is visually better to present sex and lens status as proportions using percentages instead of solely reporting absolute numbers.

Response: A percentage was additionally inserted to Sex and Lens status according to the reviewer’s comment (4 (17%) / 20 (83%), 21 (88%) /3 (12%); Results, Table 1).

o Table 1: It should be mentioned what the measures used for the symptom period are i.e. (mean +-SD, range).

Response: Type of the measures used was added as the expression of “(median (interquartile range); range)” according to the reviewer’s comment (Results, Table 1).

o Table 1: A typo in the word "months" (misspelled as "month") next to “symptom period” should be corrected.

Response: The typo in the word “months” was corrected according to the reviewer’s 

---

## [Decision Letter · Decision Letter 1]

10 Nov 2024

PONE-D-24-31532R1Topographic Changes in Macula and Its Association with Visual Outcomes in Idiopathic Epiretinal Membrane SurgeryPLOS ONE

Dear Dr. Byon,

Thank you for submitting your manuscript to PLOS ONE. After careful consideration, we feel that it has merit but does not fully meet PLOS ONE’s publication criteria as it currently stands. Therefore, we invite you to submit a revised version of the manuscript that addresses the points raised during the review process.

We look forward to receiving your revised manuscript.

Kind regards,

Kumar Saurabh

Academic Editor

PLOS ONE

Journal Requirements:

Additional Editor Comments :

This is a well executed study. Authors are invited to respond to the reviewer comments before the manuscript could be considered for acceptance.

Reviewers' comments:

Reviewer's Responses to Questions

**Comments to the Author**

1. If the authors have adequately addressed your comments raised in a previous round of review and you feel that this manuscript is now acceptable for publication, you may indicate that here to bypass the “Comments to the Author” section, enter your conflict of interest statement in the “Confidential to Editor” section, and submit your "Accept" recommendation.

Reviewer #1: All comments have been addressed

Reviewer #2: All comments have been addressed

2. Is the manuscript technically sound, and do the data support the conclusions?

Reviewer #1: Yes

Reviewer #2: Yes

3. Has the statistical analysis been performed appropriately and rigorously? 

Reviewer #1: Yes

Reviewer #2: Yes

4. Have the authors made all data underlying the findings in their manuscript fully available?

Reviewer #1: Yes

Reviewer #2: Yes

5. Is the manuscript presented in an intelligible fashion and written in standard English?

Reviewer #1: Yes

Reviewer #2: Yes

6. Review Comments to the Author

Reviewer #1: Reviewer comments:

1. The following sentence included in the Introduction: “A yellow spot on the fovea was also noted in chronic ERM fundus photograph” ; unsure what this means? Or what it means for your project? Would either rephrase/elaborate on it more or take it out all together

2. In the Figure 2 legend would change the following phrase: “Most of the movement in median foveal location tends to be occurred within 1 month” to “Most of the movement in median foveal locationtends to occur within 1 month”. It is currently grammatically incorrect

3. I would address the fact that 88% of pts had concurrent cataract surgery and discuss that this could or could not affect the results in BCVA for these pts compared to the ones that did not have cataract extraction

Reviewer #2: Great job overall- just a few minor issues/suggestions to point out:

Introduction: In the sentence "Previously studied methodologies were hard to figure out confounding variables that might affect prognosis of surgery ", replace " were hard to figure out" with "failed to identify"

Results, figure 2 legend: (B) replace "to be occured" in "Most of the movement in median foveal location

tends to be occurred within 1 month" with "occur"

Figure 1 (e): typo in "supeor"; should be "superior"

Results, figure 3 legend: the word "red" should be changed to green since the color of the dot at the disc was changed.

Results, figure 3 legend: (A): the center of the fovea was marked with a blue dot on the OCTA (not a blue cross). Same comment for (B); the fovea was marked by a red dot (not a red cross)

Discussion: Replace "in particular" with "particularly" in the sentence "The conflicting outcomes of the association between changes in macular topography and visual gain would result from the different methodologies, in particular, the extent of the macula covered by the parameters" and remove the comma after it.

Discussion: In the sentence "When the ERM contracts the macula centripetally, the retinal tissues become packed and thickened within the small, contracted macula", replace "the retinal tissues" with "the retinal layers" and replace the word "packed" with "distorted" or "wrinkled"

7. PLOS authors have the option to publish the peer review history of their article (what does this mean?). If published, this will include your full peer review and any attached files.

Reviewer #1: No

Reviewer #2: No

---

## [Author Response · Author response to Decision Letter 1]

19 Nov 2024

Dear Editor & Reviewers,

We appreciate all suggestions and corrections. All authors have carefully reviewed these comments. Our detailed responses to comments are addressed below.

Review Comments to the Author

Reviewer #1: Notes:

1. The following sentence included in the Introduction: “A yellow spot on the fovea was also noted in chronic ERM fundus photograph” ; unsure what this means? Or what it means for your project? Would either rephrase/elaborate on it more or take it out all together

 Response: To characterize the funduscopic features of the epiretinal membrane in the chronic phase, we described "yellow spot on the fovea" as subfoveal pigmentary change. However, this sentence doesn't have any relevance to our study. Therefore, based on the reviewer's recommendation, we deleted it. (Reference: Subfoveal pigment changes in patients with longstanding epiretinal membranes, Gomes NL, et. al., Am J Ophthalmol. 2009;147(5):865-868.) (Fifth sentence of second paragraph in the introduction).

2. In the Figure 2 legend would change the following phrase: “Most of the movement in median foveal location tends to be occurred within 1 month” to “Most of the movement in median foveal location tends to occur within 1 month”. It is currently grammatically incorrect

Response: We appreciate the reviewer for pointing out the grammatical error and have corrected it from “Most of the movement in median foveal location tends to be occurred within 1 month” to “Most of the movement in median foveal location tends to occur within 1 month”. (Second sentence of figure 2 legend)

3. I would address the fact that 88% of pts had concurrent cataract surgery and discuss that this could or could not affect the results in BCVA for these pts compared to the ones that did not have cataract extraction

Response: In our discussion, we described limitations cataracts can act as confounding factors as follow: “Although eyes with visually affecting cataract (more than grade 2 nuclear opalescence or cortical cataract) were excluded, mild cataract could be a potential confounding factor in assessing visual acuity.”. In order to provide a supplementary explanation for the limited impact of these limitations on this study, the results showing that opacity below grade 2 in the LOCS III system did not significantly affect visual acuity were added as follows. “However, according to the results of a study on the relationship between nuclear opalescence based on the LOCS III system and visual acuity, nuclear opalescence of grade 2 or lower had little effect on visual acuity decrease. Additionally, cortical cataracts of grade 2 or lower involve only peripheral changes and are unlikely to affect central vision. For these reason, although 88% of the patients in our study showed low-grade cataract changes, it is thought that cataract changes in the eyes included in our study would have had little effect on vision.”

(Fourth sentence of the last paragraph (limitation), Discussion).

Reviewer #2: 

1. Introduction: In the sentence "Previously studied methodologies were hard to figure out confounding variables that might affect prognosis of surgery ", replace " were hard to figure out" with "failed to identify"

Response: We appreciate the reviewer for pointing out the grammatical error and ambiguous expressions. The corresponding sentence has been corrected according to the reviewer’s comment (Third sentence of 3rd paragraph, Introduction).

2. Results, figure 2 legend: (B) replace "to be occured" in "Most of the movement in median foveal location

tends to be occurred within 1 month" with "occur"

Response: We have corrected it from “Most of the movement in median foveal location tends to be occurred within 1 month” to “Most of the movement in median foveal location tends to occur within 1 month”. (Second sentence of figure 2 legend)

3. Figure 1 (e): typo in "supeor"; should be "superior"

Response: The typo in “supeor” has been corrected to “superior” on (e) panel of figure 1.

4. Results, figure 3 legend: the word "red" should be changed to green since the color of the dot at the disc was changed. 

Response: After changing the color of the indicator in figure 3, the legend could not be modified in the first revision. According to the reviewer's comment, the red dot on the disc was changed to a green dot.

5. Results, figure 3 legend: (A): the center of the fovea was marked with a blue dot on the OCTA (not a blue cross). Same comment for (B); the fovea was marked by a red dot (not a red cross)

Response: After changing the color of the indicator in figure 3, the legend could not be modified in the first revision. According to the reviewer's comment, the blue cross and red cross in (A) and (B) were changed to blue dot and red dot.

6. Discussion: Replace "in particular" with "particularly" in the sentence "The conflicting outcomes of the association between changes in macular topography and visual gain would result from the different methodologies, in particular, the extent of the macula covered by the parameters" and remove the comma after it.

Response: The pointed out parts have been corrected as follows. “The conflicting outcomes of the association between changes in macular topography and visual gain would result from the different methodologies, particularly the extent of the macula covered by the parameters.” (Seventh sentence of 3rd paragraph, Discussion).

7. Discussion: In the sentence "When the ERM contracts the macula centripetally, the retinal tissues become packed and thickened within the small, contracted macula", replace "the retinal tissues" with "the retinal layers" and replace the word "packed" with "distorted" or "wrinkled"

Response: The pointed out parts have been corrected as follows. “When the ERM contracts the macula centripetally, the retinal layers become distorted and thickened within the small, contracted macula” (Seventh sentence of 5th paragraph, Discussion).

---

## [Decision Letter · Decision Letter 2]

17 Dec 2024

Topographic Changes in Macula and Its Association with Visual Outcomes in Idiopathic Epiretinal Membrane Surgery

PONE-D-24-31532R2

Dear Dr. Byon,

We’re pleased to inform you that your manuscript has been judged scientifically suitable for publication and will be formally accepted for publication once it meets all outstanding technical requirements.

Kind regards,

Kumar Saurabh

Academic Editor

PLOS ONE

Additional Editor Comments (optional):

Dear Authors

Please find the reviewers comment about grammatical errors. Manuscript may be acceptable post revision.

Reviewers' comments:

Reviewer's Responses to Questions

**Comments to the Author**

1. If the authors have adequately addressed your comments raised in a previous round of review and you feel that this manuscript is now acceptable for publication, you may indicate that here to bypass the “Comments to the Author” section, enter your conflict of interest statement in the “Confidential to Editor” section, and submit your "Accept" recommendation.

Reviewer #1: All comments have been addressed

Reviewer #2: All comments have been addressed

2. Is the manuscript technically sound, and do the data support the conclusions?

Reviewer #1: Yes

Reviewer #2: Yes

3. Has the statistical analysis been performed appropriately and rigorously? 

Reviewer #1: Yes

Reviewer #2: Yes

4. Have the authors made all data underlying the findings in their manuscript fully available?

Reviewer #1: Yes

Reviewer #2: Yes

5. Is the manuscript presented in an intelligible fashion and written in standard English?

Reviewer #1: Yes

Reviewer #2: Yes

6. Review Comments to the Author

Reviewer #1: all comments made by all reviewers have been adequately addressed by the authors and corrections have been made as needed

Reviewer #2: Great job. Just a couple of minor comments:

In the introduction: remove "were" from the sentence: "Previously studied methodologies were failed to identify hard to figure out confounding variables that might affect prognosis of surgery [13,15]."

In the introduction: add "an" to : "Serial fundus photographs showed both the membrane reflex on the macula

in the early stage and the wrinkled, opaque macula with decreased distance between

vascular arcades at (an) advanced stage."

7. PLOS authors have the option to publish the peer review history of their article (what does this mean?). If published, this will include your full peer review and any attached files.

Reviewer #1: No

Reviewer #2: No

---

## [Editor Report · Acceptance letter]

30 Dec 2024

PONE-D-24-31532R2 

PLOS ONE

Dear Dr. Byon, 

I'm pleased to inform you that your manuscript has been deemed suitable for publication in PLOS ONE. Congratulations! Your manuscript is now being handed over to our production team.

Kind regards, 

on behalf of

Dr. Kumar Saurabh 

Academic Editor

PLOS ONE